# Improvement of Cell Culture Methods for the Successful Generation of Human Keratinocyte Primary Cell Cultures Using EGF-Loaded Nanostructured Lipid Carriers

**DOI:** 10.3390/biomedicines9111634

**Published:** 2021-11-06

**Authors:** Jesús Chato-Astrain, David Sánchez-Porras, Óscar Darío García-García, Claudia Vairo, María Villar-Vidal, Silvia Villullas, Indalecio Sánchez-Montesinos, Fernando Campos, Ingrid Garzón, Miguel Alaminos

**Affiliations:** 1Tissue Engineering Group, Department of Histology, Universidad de Granada, 18016 Granada, Spain; jchato@ugr.es (J.C.-A.); david.s.p.94@gmail.com (D.S.-P.); e.oscargg@go.ugr.es (Ó.D.G.-G.); fcampos@ugr.es (F.C.); 2Instituto de Investigación Biosanitaria ibs.GRANADA, 18012 Granada, Spain; ismg@ugr.es; 3BioKeralty Research Institute AIE, Albert Einstein, 25-E3, 01510 Miñano, Spain; claudia.vairo@keralty.com (C.V.); silvia.villullas@bioaraba.org (S.V.); 4Keralty Health SI, Barrio Arkauti, 5, 01192 Vitoria-Gasteiz, Spain; maria.villar@keralty.com; 5Department of Human Anatomy and Embryology, University of Granada, 18016 Granada, Spain

**Keywords:** EGF, nanoparticles, NLC, keratinocytes, tissue engineering, cell culture

## Abstract

Human skin keratinocyte primary cultures can be established from skin biopsies with culture media containing epithelial growth factor (EGF). Although current methods are efficient, optimization is required to accelerate the procedure and obtain these cultures in less time. In the present study, we evaluated the effect of novel formulations based on EGF-loaded nanostructured lipid carriers (NLC). First, biosafety of NLC containing recombinant human EGF (NLC-rhEGF) was verified in immortalized skin keratinocytes and cornea epithelial cells, and in two epithelial cancer cell lines, by quantifying free DNA released to the culture medium. Then we established primary cell cultures of human skin keratinocytes with basal culture media (BM) and BM supplemented with NLC-rhEGF, liquid EGF (L-rhEGF), or NLC alone (NLC-blank). The results showed that cells isolated by enzymatic digestion and cultured with or without a feeder layer had a similar growth rate regardless of the medium used. However, the explant technique showed higher efficiency when NLC-rhEGF culture medium was used, compared to BM, L-rhEGF, or NLC-blank. Gene expression analysis showed that NLC-rhEGF was able to increase *EGFR* gene expression, along with that of other genes related to cytokeratins, cell–cell junctions, and keratinocyte maturation and differentiation. In summary, these results support the use of NLC-rhEGF to improve the efficiency of explant-based methods in the efficient generation of human keratinocyte primary cell cultures for tissue engineering use.

## 1. Introduction

Advanced therapy medicinal products (ATMP) are biomedicines for human use based on genes, tissues, or cells, which offer groundbreaking new opportunities for the treatment of diseases and injuries [1]. Although the availability of advanced therapies has significantly increased in recent years, only a few ATMP are currently available to patients, particularly in the case of tissue-engineered products [1]. One of the ATMP approved for clinical use in Spain is the UGRSKIN bioartificial skin model generated in the laboratory for patients affected by large burns [2]. Like most full-thickness biological substitutes of human skin [3], UGRSKIN consists of an epidermal layer cultured on the surface of a dermal substitute, and requires the establishment of stromal and epithelial cell cultures [4].

Several methods have been described for the efficient generation of skin keratinocyte cell cultures [5]. However, human keratinocytes typically show low proliferation rates, and establishing abundant keratinocyte cultures usually requires prolonged periods of time, because epithelial stem cells are slow-cycling cells [6]. One of the main challenges in the treatment of patients with severe burns is to achieve rapid, effective closure of the skin injuries to prevent fluid loss and nosocomial infections, the two major complications in these patients [7,8,9]. For this reason, further research focused on shortening the time required to fabricate bioartificial skin is needed.

In this connection, several growth factors, cytokines, chemokines, and other bioactive molecules are known to be successful accelerators of cell proliferation and to favor keratinocyte expansion ex vivo when used to supplement the culture medium [3,10]. Several researchers previously suggested that the use of histamine and vitamin D can help increase cell proliferation, in light of different findings that demonstrated the role of these components in keratinocyte proliferation [11,12]. Among bioactive molecules that have shown potential to increase keratinocyte activation, proliferation, attachment, and motility, the most favorable candidate to date appears to be epidermal growth factor (EGF) [10,13], which is used in most keratinocyte culture protocols. In fact, EGF is known to stimulate keratinocyte growth and improve wound healing, and is necessary for tissue regeneration [14,15]. However, the half-life of EGF is brief, and this molecule can be disrupted and degraded after in vivo administration [16]. Therefore, achieving rapid keratinocyte expansion in culture remains the major limiting step in the generation of bioartificial skin.

Novel nanomedicine-based formulations can enhance EGF stability and allow the controlled release of this bioactive factor. Nanostructured lipid carriers (NLC) were described as promising second-generation lipid-based delivery systems capable of controlling EGF release, increasing its effectiveness, and optimizing its administration in terms of dose, delivery pattern, and safety [15,17,18,19,20]. Compared to similar technologies, such as the use of solid lipid nanoparticles, NLC offer increased loading capacity and efficiency [21], along with faster drug release [22]. In addition, NLC are more stable, making them easier to use than other types of particles [22]. Moreover, the NLC used in the present study are able to protect the bioactive compound associated to the nanocarrier from hydrolysis and oxidation, and provide controlled release [23]. A further advantage of NLC is that they can be administered topically on the skin, which facilitates their use in treating skin ulcers and injuries [21,22]. When NLC were loaded with EGF, these carriers demonstrated a clear capacity to improve skin regeneration when applied topically to a lesion in porcine [17] and mouse models [15]. In both cases, application of NLC containing EGF was more effective than free EGF, and was able to induce keratinocyte proliferation and wound regeneration. However, the effect of EGF-loaded NLC has not yet been determined in primary cell cultures of human skin keratinocytes established from human skin biopsy samples.

In the present study we evaluated the potential of human recombinant EGF (rhEGF)-loaded NLC to improve methods currently available to generate primary cultures of human skin keratinocytes, with the ultimate goal of shortening the time required to produce an efficient skin substitute by tissue engineering.

## 2. Materials and Methods

### 2.1. Cell Cultures

For the biosafety analyses, immortalized human epithelial cells Ker-CT (CRL-4048), and corneal epithelial cells SIRC (ATCC CCL-60) were purchased from ATCC (Rockville, MD, USA) and cultured with the media recommended by the supplier. Cell lines corresponding to epidermoid carcinoma cells A431 (CRL-1555) and lung carcinoma cells A549 (CRM-CCL-185) were also obtained from ATCC and cultured as recommended.

Primary cell cultures of human skin keratinocytes (HKC) were established from small full-thickness skin biopsies obtained from healthy donors who underwent skin surgery. Immediately after excision, skin samples were kept at 4 °C in Dulbecco’s modified Eagle’s medium (DMEM; Merck, Darmstadt, Germany), supplemented with antibiotics and antimycotics (100 U/mL penicillin G, 100 mg/mL streptomycin, and 0.25 mg/mL amphotericin B; Merck), and delivered to the laboratory. Biopsy samples were then washed twice in phosphate-buffered saline (PBS), surgically prepared to remove any remaining adipose tissue, and processed with two different methods routinely used to establish keratinocyte cultures:(1)Cell isolation by enzymatic digestion. Tissue samples were incubated in trypsin-EDTA as previously reported [24]. Briefly, samples were placed in a commercial solution containing 0.05% trypsin and 0.02% EDTA (Merck) at 37 °C with gentle shaking. After 20 min, the solution was harvested and inactivated with culture medium, and detached epithelial cells were harvested from the dissociation solution by centrifugation. This procedure was repeated up to 10 times, and all harvested cells were cultured together in 6-well plates. In one of the study groups, harvested cells were cultured on a feeder layer previously cultured on the plates. For this group, 3T3 cells (Merck) were cultured on 6-well plates and a lethal dose of gamma irradiation was applied when cells reached semiconfluence. The cells were washed with PBS, and then dissociated cells were cultured directly on the surface of this cell layer.(2)Explant technique. The tissues were trimmed into small pieces with a surgical blade, and each piece was placed on the surface of a 6-well plate, with the epithelial layer in direct contact with the culture surface. These explants were allowed to attach to the surface for 30 min, and a very small amount of culture medium was carefully added to prevent explant detachment. Culture plates were placed in a cell incubator, and 5 mL of culture medium was added to each well 24 h later.

In all cases, cells were incubated at 37 °C with 5% CO_2_ under standard culture conditions and with specific culture media in each study group (see Section 2.4). The culture medium was changed every 3 days.

### 2.2. Preparation of NLC

rhEGF-loaded NLC [18,20,25,26,27] and NLC alone (NLC-blank) were prepared with the hot melt homogenization technique according to a modified protocol described by Gainza and colleagues [15]. Briefly, an aqueous solution was prepared by adding 20 mg/mL rhEGF (Center for Genetic Engineering and Biotechnology, Havana, Cuba) to a warmed (50 °C) solution containing 0.67% (*w*/*v*) Poloxamer 188 (BASF, Ludwigshafen, Germany) and 1.33% (*w*/*v*) Tween 20 (Panreac Química, Barcelona, Spain). To prepare NLC alone, no rhEGF was added. This aqueous solution was then added to a previously melted lipid blend consisting of a 10:1 mixture of Precirol^®^ ATO 5 (Gattefossé, Madrid, Spain) and Miglyol 182 N/F (Gattefossé) and emulsified under sonication for 30 s at 50 W (Branson^®^ 250 Sonifier, Danbury, CT, USA). The resulting emulsion was first kept at room temperature for 30 min under gentle stirring and later cooled to 4 °C for 2 h to allow NLC formation. The resulting nanoparticles were freeze-dried after the addition of trehalose as a cryopreserving agent at a final concentration of 15% (*w*/*w*) of the weighed lipid. No washing step was used, therefore, both encapsulated and nonencapsulated rhEGF were present on the nanoparticle surfaces.

### 2.3. Characterization of rhEGF-Loaded NLC

rhEGF-loaded NLC were characterized in terms of size and dispersion (span) with a Nanosight NS300 Zetasizer Nano ZS instrument (Malvern Panalytical Ltd., Malvern, UK) based on nanoparticle tracking analysis (NTA). Zeta potential (ζ) was determined with a Zetasizer Nano ZS instrument (Malvern Instruments, Worcestershire, UK) based on Doppler velocimetry (LDV).

rhEGF loading was calculated with the following formula:$$rhEGF\;loading=\frac{Initial\;rhEGF\;amount\;\left(mg\right)}{Total\;batch\;weight\;\left(mg\right)}$$

Encapsulation efficiency (EE%) was estimated indirectly by quantifying the amount of free rhEGF present in the supernatant, as described by Gainza and colleagues [15]. Briefly, 5 mL of the cooled suspension was centrifuged in an Amicon^®^ centrifugal filtration unit (100 kDa molecular weight cut-off, Merck-Millipore, Burlington, MA, USA) for 15 min at 2500 rpm to determine the amount of nonencapsulated drug. The amount of rhEGF was determined with the Bradford method. Once this quantity was known, EE% was calculated with the following formula:$$EE\left(\%\right)=\left(\frac{Initial\;rhEGF\;amount-Free\;rhEGF}{Initial\;rhEGF\;amount}\right)\times 100$$

Samples were analyzed in triplicate to obtain an accurate mean value, except for drug loading.

### 2.4. Study Groups

The following study groups were used:(1)BM. In this group, cells were cultured in basic culture medium (BM) consisting of a 3:1 mixture of DMEM and Ham’s F-12 supplemented with 10% fetal bovine serum (FBS), 1% antibiotics and antimycotics, 24 µg/mL adenine, 0.4 µg/mL hydrocortisone, 5 µg/mL insulin, and 1.3 ng/mL triiodothyronine. Cells cultured in this medium were used as a control.(2)L-rhEGF. In this group, cells were cultured in BM supplemented with liquid recombinant human EGF (L-rhEGF) at a final concentration of 10 ng/mL. This medium corresponds to the epithelial medium used for keratinocyte culture and expansion in the UGRSKIN model of tissue-engineered skin.(3)NLC-rhEGF. In this group, cells were cultured with BM supplemented with rhEGF-loaded NLC (NLC-rhEGF) at a final concentration of 10 ng/mL.(4)NLC-blank. NLC alone were added to BM at the same concentration used in group 3 (10 ng/mL). This group was also considered a control, since no EGF was used.

In all cases, cells cultured with BM, L-rhEGF, NLC-rhEGF, or NLC-blank were incubated at 37 °C with 5% CO_2_ under the same culture conditions.

### 2.5. Biosafety Analysis of NLC on Epithelial Cells

To determine the effect of NLC on cell viability, the DNA quantification method was used as previously reported [28]. Briefly, human epithelial Ker-CT cells, SIRC corneal epithelial cells, and the cell lines A431 and A549 were subcultured on 24-well plates. Twenty-four hours later, the culture medium was replaced with BM, L-rhEGF, NLC-rhEGF, or NLC-blank, and cells were incubated for 10 days under standard culture conditions. Samples from the culture supernatant were taken after 1, 4, 7, and 10 days of culture in each condition, and DNA released from cultured cells was quantified by determining absorbance at 260/280 nm with a NanoDrop 2000 spectrophotometer (Thermo Fisher Scientific, Waltham, MA, USA). These experiments were carried out in quintuplicate (*n* = 5).

### 2.6. Proliferation Potential of Each Culture Medium

To determine the potential of each type of culture medium to favor the successful generation of primary cell cultures, skin biopsy samples were processed by (1) enzymatic digestion and cell culture without a feeder layer, (2) enzymatic digestion and cell culture with a feeder layer, and (3) the explant technique. In all three cases, cells were cultured in the presence of each type of culture medium (BM, L-rhEGF, NLC-rhEGF, or NLC-blank). For cells processed with the enzymatic digestion protocol, images were obtained after 3, 7, 11, and 14 days of follow-up with an inverted phase-contrast microscope, and the number of HKC found at each time in each sample was quantified. In tissue samples processed with the explant method, we quantified the percentage of explants that succeeded in generating keratinocyte colonies, fibroblast colonies, and mixed keratinocyte–fibroblast colonies. All experiments were carried out in quintuplicate (*n* = 5).

### 2.7. Gene Expression Analysis

To determine the effect of each type of culture medium on gene expression, human keratinocytes cultured with BM, L-rhEGF or NLC-rhEGF were analyzed by RT-PCR. First, total RNA was extracted from each type of culture with a Qiagen RNeasy Mini Kit (Qiagen, Mississauga, ON, Canada). Then RNA was retrotranscribed to cDNA with an iScript Advanced cDNA Synthesis Kit for RT-qPCR (Bio-Rad Laboratories, Hercules, CA, USA). The cDNA was used as a template to amplify and quantify the different genes analyzed here with a customized PCR plate in the Prime-PCR system (Bio-Rad Laboratories). This plate contained 21 genes associated with epithelial differentiation and function, and several control genes related with keratinization (cytokeratins 1, 5, 6A, 7, 8, 10, 13, 16, and 19), keratinocyte differentiation (filaggrin and involucrin), cell–cell junction development (demosome proteins DSC1, DSG1, DSP, PKP1, and PPL; GAP junction proteins GJA1 and GJA4; and tight junction proteins TJP1 and TJP2), and the expression of the EGF receptor gene (EGFR). Briefly, equal amounts of cDNA were mixed with SsoAdvanced Universal SYBR^®^ Green Supermix (Bio-Rad Laboratories) and the PCR reaction was carried out according to the manufacturer’s instructions in a Bio-Rad CFX Connect 96 thermocycler. All results were normalized to GAPDH expression, and the fold-change (FC) in expression in each study group was determined with cells cultured in BM as a control (expression = 1) with Bio-Rad CFX Manager software (Bio-Rad Laboratories). All RNA expression analyses were carried out in duplicate (*n* = 2).

### 2.8. Statistical Analysis

The results of quantitative analyses of HKC cultured with the enzymatic digestion protocol (with and without a feeder layer) and the explant protocol were compared in the different study groups (BM, L-rhEGF, NLC-rhEGF, or NLC-blank) with the Mann–Whitney nonparametric test. To determine the correlation between two variables, we used the Kendall tau correlation test. All comparisons were performed with Real Statistics Resource Pack software (Release 7.2) (Dr. Charles Zaiontz, Purdue University, West Lafayette, IN, USA), available at www.real-statistics.com (access date: 5 November 2021).

## 3. Results

### 3.1. rhEGF-Loaded NLC Characterization

The characteristics of the NLC are summarized in Table 1. Dispersion analysis confirmed satisfactory particle homogeneity for both samples (span < 1.2). In addition, the rhEGF loading obtained was 0.04 mg/mg, and the EE% was 92.45 ± 4.6%.

### 3.2. Biosafety Analysis of NLC in Epithelial Cell Lines

The preliminary cell viability analysis showed that the amount of DNA released by each cell line cultured in the different culture media with or without NLC was very low, and never exceeded the basal levels found in control cells cultured in BM without NLC (Appendix A). These findings suggested that the use of culture media enriched with L-rhEGF, NLC-rhEGF, or NLC-blank did not reduce cell viability compared to BM.

### 3.3. Establishment of Primary Cell Cultures of HKC with the Cell Isolation Technique

Analysis of isolated HKC cultured on cell culture plates without a feeder layer showed progressive coating of the culture surface with time. As shown in Figure 1, cells were scattered over the surface on day 3 of culture and tended to spread and cover the entire surface by day 14. Quantification of the culture surface area occupied by HKC confirmed the steady increase in cells at each time point, with a significant correlation between area and follow-up time (*r* = 0.6965; *p* < 0.001 for the Kendall tau correlation test). Regarding the different culture media, we found that the area covered by cultured cells tended to be greater in the BM+L-EGF and BM+NP-EGF groups than in the BM and BM+NP-blank groups on days 7, 11, and 14, but the differences were not statistically significant (*p* > 0.05).

When isolated HKC were cultured with a feeder layer, we also observed progressive coating of the culture surface (although the surface at the initial time was already covered by the feeder layer), with a significant correlation between surface area and the duration of culture (*r* = −0.2052; *p* = 0.0152 for the Kendall tau correlation test). As shown in Figure 2, the area covered by skin cells tended to increase with time. As in cultures prepared with no feeder layer, the highest values were found for cells cultured with BM+L-EGF and BM+NP-EGF media, although the differences among groups were not statistically significant (*p* > 0.05).

### 3.4. Establishment of Primary Cell Cultures of HKC with the Explant Technique

We analyzed the capability of each cell culture medium to favor the successful establishment of skin cell cultures with the explant technique (Figure 3). The results showed significant differences among the different study groups. Specifically, BM+NP-EGF was the best medium to establish primary keratinocyte cultures from explants, with 30.12 ± 2.06% of the explants yielding pure keratinocyte cultures. This percentage was significantly larger (*p* = 0.0285) than in the other three conditions compared here (12.28 ± 2.16% for BM, 16.58 ± 1.85% for BM+L-EGF, and 8.55 ± 2.36% for BM+NP-blank). Interestingly, culture success in the BM+L-EGF group was significantly greater than in the BM+NP-blank group (*p* = 0.0285). Analysis of the percentage of explants that produced fibroblast growth also disclosed a significantly better success rate in the BM+NP-EGF group, with an EE% of 20.71 ± 2.86% compared to 13.42 ± 3.08% in the BM group, 8.44 ± 2.35% in the BM+L-EGF group, and 7.63 ± 2.06% in the BM+NP-blank group. Interestingly, we also found that some explants were able to generate mixed cultures containing both keratinocytes and fibroblasts. We found that 16.71 ± 2.21% of the explants cultured with BM generated a mixed culture, whereas in the BM+L-EGF group 12.39 ± 3.14% of the explants gave rise to mixed cultures. In the BM+NP-EGF group, mixed cultures were seen for 7.52 ± 2.58% of the explants, and in the BM+NP-blank group, these cultures were seen for 8.35 ± 2.19% of the explants. In these last two groups, the percentages were significantly lower than in the BM group. A final observation was that the percentage of explants that failed to generate any cell cultures differed among the study groups, with 57.59 ± 5.71% of unsuccessful explants in the BM group, 62.58 ± 5.28% in BM+L-EGF, 41.65 ± 5.26% in BM+NP-EGF, and 75.46 ± 6.35% in the BM+NP-blank group. In the BM+NP-EGF group this percentage was significantly lower than in the other three groups, whereas the percentage value in the BM+NP-blank group was significantly higher than in the remaining three groups.

### 3.5. Gene Expression Analysis

Our preliminary analysis of gene expression in cells cultured under different conditions disclosed an inductive effect of the nanoparticles used in this study on several types of genes (Figure 4). The *EGFR* gene, which encodes the EGF receptor, was induced in the BM+NP-EGF group, and this effect was more than 7-fold greater than in cells cultured in BM, whereas the use of BM+L-EGF was associated with a 1.68-fold increase in expression.

Several other genes related to epithelial cell differentiation, including cytokeratins, filaggrin, and involucrin, were also analyzed. Cytokeratins 1, 5, 7, 8, 13, 16, and 19, filaggrin, and involucrin showed a more than 2-fold increase in the BM+NP-EGF group, with lower expression in the BM+L-EGF group. Similarly, we found that several genes related to the development of different types of intercellular junctions were overexpressed upon induction with BM+NP-EGF (more than 2-fold), including the desmosome proteins DSC1, DSG1, DSP, PKP1, and PPL, the GAP junction proteins GJA1 and GJA4, and the tight junction proteins TJP1 and TJP2. Some of these genes were also induced by BM+L-EGF, albeit at lower levels than in the BM+NP-EGF group.

## 4. Discussion

Since publication of the first description of a culture technique to grow human keratinocytes [29], several methods have been described to efficiently culture human epithelial cells to produce ATMP. Application of these methods currently makes it possible to establish primary cell cultures from a small skin biopsy sample, which, in turn, allows bioartificial substitutes for human skin to be generated by tissue engineering in 3–4 weeks [2,4,30]. However, this timeframe may be too long when treating patients with severe burns. Therefore, novel methods able to reduce this time are needed.

In the present study, we evaluated the effect of NLC-rhEGF, and found that these compounds can contribute to the efficient development of human keratinocyte cultures. First, we confirmed that these nanoparticles fulfilled the basic biosafety requirements for use in cell culture, since all cell types evaluated here displayed high cell viability, and cell survival was not affected by NLC-rhEGF treatment. These results are in agreement with previous findings that these nanoparticles are safe and can be used without compromising cell viability [16], as found previously for other types of NLC [20,25,26]. Compared to previously described NLC, the physicochemical properties of our carriers showed some interesting differences, such as particle size, which was smaller in our case than in previous reports [15,17,31], probably due to slight differences in the synthesis protocols. Although additional work is needed to demonstrate the effect of particle size on biological function, the fact that our NLC were smaller is a potentially positive feature, since it has been demonstrated that effective cell uptake and permeability of this type of particle are directly related to particle size [32]. In addition, we found that NLC loaded with rhEGF were larger than blank NLC, even though the same method was used to generate both types of particle. Although the reasons for this difference are not known, we hypothesize that the incorporation of rhEGF is associated with a slight modification in the crystalline structure of the particles, which has been shown to play a role in controlling particle size [33]. Other important parameters such as particle span, size dispersion, and zeta potential were similar to previous reports, suggesting that the biofunctionality and stability of our particles are likely to be similar to those of previously tested NLC [15,17,31]. As previously reported, our NLC allowed for a sustained release of EGF in water, which allowed for a dose reduction [15,31].

Next, we compared the pro-proliferative effects of NLC-rhEGF vs. L-rhEGF in primary cultures of human skin keratinocytes. Two main methods have been described to generate skin keratinocyte cultures: cell isolation by enzymatic digestion, and the explant technique [5]. Enzymatic digestion is based on the use of collagenase, trypsin, dispase, or other enzymes able to disaggregate tissue biopsy samples to release cells in a suspension [34,35,36]. Suspended cells can then be harvested by centrifugation, and cultured in flasks alone or in the presence of an inactivated feeder layer consisting of previously cultured 3T3 mouse fibroblasts [36,37], as originally proposed by Rheinwald and Green [29]. Although feeder layers are able to increase the efficiency of the isolation method, the xenogeneic origin of these cells makes it necessary to search for alternative methods free of these mouse cells. In this connection, NLC-rhEGF was no more efficient than L-rhEGF in promoting cell proliferation in human primary keratinocyte cultures obtained by enzymatic digestion, even when a feeder layer was used. Previous research demonstrated that NLC-rhEGF have an notable mitogenic effect on HaCaT keratinocytes and fibroblasts, and proliferation was significantly increased upon culture with NLC-rhEGF [15]. Although differences between our results and previous findings are difficult to explain, we hypothesize that primary cell cultures of normal keratinocytes behave in a different manner than immortalized cell lines, which have a higher rate of cell proliferation and may, therefore, be more sensitive to NLC-rhEGF. In addition, the previous results cited above used low concentrations of serum, whereas the cells used in the present study were cultured and maintained with high concentrations of bovine serum to favor cell proliferation, given that low concentrations of serum can inhibit cell proliferation in culture [38].

In addition, we carried out gene expression analysis of human keratinocytes subjected to enzymatic digestion by quantifying the expression of several relevant genes in these cell cultures. Interestingly, we found that NLC-rhEGF was indeed capable of increasing *EGFR* gene expression compared to L-rhEGF. The *EGFR* gene encodes a receptor that is activated by EGF and other proteins, and its function is related to cell proliferation, but also to cell attachment, motility and differentiation, along with other cell functions [39]. The fact that *EGFR* was significantly induced by NLC-rhEGF with no increase in cell proliferation suggests that NLC-rhEGF may efficiently induce the EGF receptor, but the effects of this induction are more closely related to the nonproliferative effects of *EGFR* rather than to cell proliferation. In this regard, it is well known that *EGFR* activation is able to promote terminal keratinocyte differentiation and keratinization [39], which is in line with our finding that culture with NLC-rhEGF was associated with the induction of several genes related to cytokeratins, intercellular junctions, and other relevant epithelial cell genes. As noted above, the differences between our results in primary cell cultures and previous findings in immortalized cell lines require further research, and future experiments should be carried out with primary cultures of normal keratinocytes.

Explant techniques, on the other hand, are based on the use of small tissue biopsy samples that are placed directly on culture surfaces and then grown with specific media [40]. With these methods, cells are able to directly migrate from the edges of the tissue explant, adhere, and multiply on the culture surface [41]. In general, the efficiency of this method is high. Nonetheless, correct placement and attachment of the tissue explant to the culture surface have been shown to be crucial for the success of the technique [41]. Although the explant method has not been extensively used in skin keratinocyte cultures, several reports suggest that explants may be more efficient than enzyme digestion to establish primary cultures of keratinocytes from other tissues such as the human oral mucosa [42,43]. Compared to the enzymatic digestion procedure, the explant technique offers the advantage of requiring only very small tissue fragments to generate primary cell cultures, despite the longer time required [42]. Interestingly, our results suggest that NLC-rhEGF may have a positive effect on normal human keratinocytes obtained from human skin explants, with significantly higher cell yields generated from each tissue explant compared to other culture techniques. The reasons why NLC-rhEGF was efficient in this case remain elusive and need further research to elucidate. However, NLC-rhEGF was previously shown to be capable of inducing skin healing and keratinocyte proliferation in vivo, and putative effectiveness has been suggested for the treatment of chronic wounds [15,17]. The fact that skin explants more closely reproduce in vivo conditions than isolated cells subjected to enzymatic digestion may help explain our results. In this connection, the positive effect of NLC-rhEGF compared to L-rhEGF may be mediated by the action of stromal cells, extracellular matrix components, or other molecules that are present in the explants, but not in isolated cells.

To our knowledge, this is one of the first reports of a method able to reduce the time needed to generate primary keratinocyte cell cultures with the explant technique. Given the advantages of explant methods compared to enzymatic digestion, our results support the application of the former approach in protocols designed to isolate primary cultures of human keratinocytes for the generation of ATMP by tissue engineering. The use of NLC-rhEGF can increase the potential of explant techniques to more rapidly generate keratinocyte cultures for future clinical use; thus, the present findings confirm the putative usefulness of NLC-based technologies, as previously proposed [18,20,25,26,27].

## Figures and Tables

**Figure 1 biomedicines-09-01634-f001:**
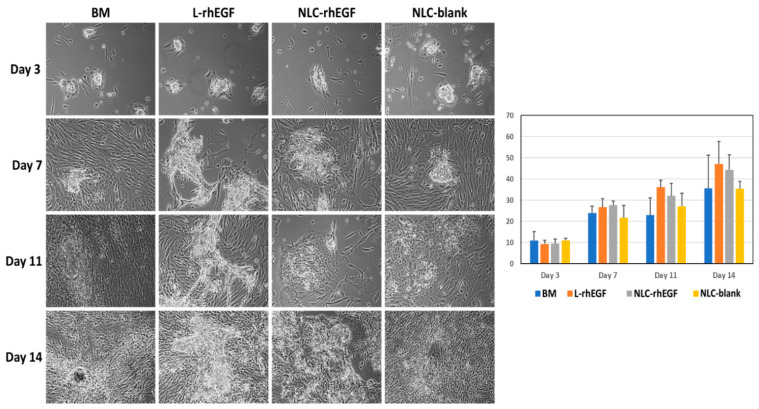
Surface area occupied by HKC cultured on flasks without a feeder layer at different follow-up times (days 3, 7, 11, and 14 of culture) in the presence of four different culture media. The histogram on the right shows the quantification analysis of surface areas in each study group as average values, with error bars for standard deviations. Images were taken with 100× magnification.

**Figure 2 biomedicines-09-01634-f002:**
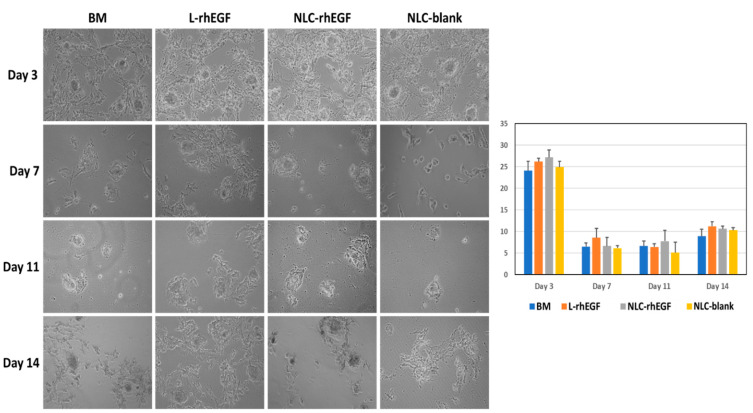
Surface area occupied by HKC cultured on flasks with a feeder layer at different follow-up times (days 3, 7, 11, and 14 of culture) in the presence of four different culture media. The histogram on the right shows the quantification analysis of surface area in each study group as average values, with error bars for standard deviations. Images were taken with 100× magnification.

**Figure 3 biomedicines-09-01634-f003:**
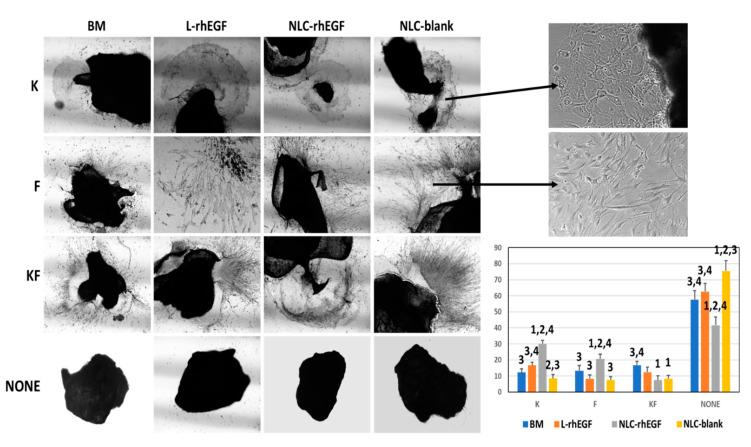
Establishment of primary cell cultures of human skin cells with the explant technique. For each culture medium, a representative image is shown for explants that produced keratinocyte colonies (K), fibroblast colonies (F), and mixed keratinocyte–fibroblast cells (KF), and for explants that failed to generate any type of cell colony (NONE) using 40× magnification. Histological images on the upper right at higher magnification (200×) illustrate a keratinocyte colony and fibroblast cells arising from the explants, and the histogram on the lower right shows the percentage of explants that generated each type of colony. 1: significantly different from the BM group; 2: significantly different from the BM+L-EGF group; 3: significantly different from the BM+NP-EGF group; and 4: significantly different from the BM+NP-blank group.

**Figure 4 biomedicines-09-01634-f004:**
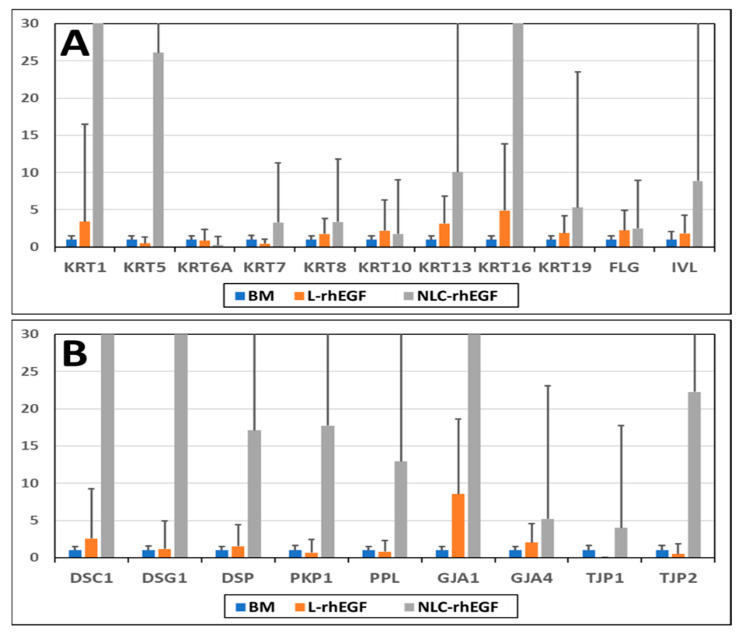
Gene expression analysis as determined by RT-PCR of cells cultured under different conditions. The results are shown as fold-change in RNA expression in each study group after normalization to cells cultured in BM as the baseline value (expression = 1). (**A**): results for several cytokeratins, filaggrin and involucrin; (**B**): results for several intercellular junctions.

**Table 1 biomedicines-09-01634-t001:** Physicochemical characterization of rhEGF-loaded NLC and NLC-blank. Nanoparticle mean size, span, and zeta potential are shown. Values are means ± standard deviations.

Formulation	Size (nm)	Span	Zeta Potential (mV)
rhEGF-NLC	137 ± 3.4	0.7	−32 ± 0.31
NLC-blank	112 ± 5.2	0.9	−33 ± 0.42

## Data Availability

The data presented in this study are available on request from the corresponding author.

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
