# Peer review of "Improvement of Cell Culture Methods for the Successful Generation of Human Keratinocyte Primary Cell Cultures Using EGF-Loaded Nanostructured Lipid Carriers"

_biomedicines, 2021, doi:10.3390/biomedicines9111634_

Round 1
Reviewer 1 Report
The topic submitted is novel and adds significant research data to the existing field of research. The article is not very well articulated needs English language revisions and even formatting of the manuscript as per the MDPI guidelines. The manuscript needs to be checked for statistical significance and clear figures. Can be accepted after minor revisions. Authors are advised to add relevant references of nanostructured lipid carriers and their role in redesigning EGF delivery systems. Below are a few suggestions of articles to be added in results, discussion, and methodology.
International journal of pharmaceutics 569 (2019): 118484
Journal of liposome research 26, no. 4 (2016): 288-296.
Scientific reports, 10(1), pp.1-15.
Gels 7, no. 3 (2021): 96.
Cancer Translational Medicine 2, no. 5 (2016).
Author Response
AUTHORS’ RESPONSE: Thank you very much for your kind comments.
The manuscript has been carefully revised and checked by a professional native English-speaker editor, as suggested. In addition, we have included all the referred references in the revised version of the manuscript.
Reviewer 2 Report
In this study, authors have examined the potential of human recombinant EGF (rhEGF)-loaded nanostructured lipid carriers and their possibility to improve already used methods in order to generate primary cultures of human skin keratinocytes. Their main goal was to produce an efficient skin substitute by tissue. The topic of the work is very interesting and significant since advanced therapy medicinal products can offer great opportunities for the treatment of diseases and injuries. From my point of view, article can be accepted after some minor comments; besides, limiting the time needed to generate primary keratinocyte cell cultures is on high demand.
Comment 1. Given that various nanocarriers based on NLC loaded with EGF, have been already produced, please state the novelty of your study in the introduction. You can a paragraph discussing specific papers with EGF-loaded NLC.
Comment 2. Please check for syntax and grammar errors
Comment 3. Compare your physicochemical results of NLC with previous published articles
Comment 4. You should provide us with in vitro release data of the EGF from the NLC, if available. Its release may also explain your gene expression data
Comment 5. Explain why the size of EGF loaded NLC is higher than the blank NLC.
Comment 6. Some sentences are too long and difficult to follow i.e. lines 378-381, please check carefully the language.
Author Response
Comment 1. Given that various nanocarriers based on NLC loaded with EGF, have been already produced, please state the novelty of your study in the introduction. You can a paragraph discussing specific papers with EGF-loaded NLC.
AUTHORS’ RESPONSE: This is a very interesting suggestion. Following the recommendations of the reviewer, we have included a paragraph in the introduction section with some new references related to the different types of NLC loaded with EGF (lines 72-82).
Comment 2. Please check for syntax and grammar errors
AUTHORS’ RESPONSE: The manuscript has been carefully revised and checked by a professional native English-speaker editor, as suggested.
Comment 3. Compare your physicochemical results of NLC with previous published articles
AUTHORS’ RESPONSE: The physico-chemical features of our NLC were compared with previously published reports. As stated in the discussion section of the revised manuscript, we found that the average size of our NLC was smaller than other types of NLC, although particle dispersion and zeta potential were similar.
Comment 4. You should provide us with in vitro release data of the EGF from the NLC, if available. Its release may also explain your gene expression data
AUTHORS’ RESPONSE: The in vitro release results of the NLC used in the present work were previously published [1,2]. In general, a sustained release of EGF was found when rhEGF-NLC were resuspended in water, which allows a EGF dose reduction and decreases the possibility of side effects [1]. This behavior allows the prolonged release of rhEGF [2]. This important information has been incorporated to the discussion section of the revised manuscript (lines 386-388).
- Gainza, G.; Pastor, M.; Aguirre, J.J.; Villullas, S.; Pedraz, J.L.; Hernandez, R.M.; Igartua, M. A Novel Strategy for the Treatment of Chronic Wounds Based on the Topical Administration of RhEGF-Loaded Lipid Nanoparticles: In Vitro Bioactivity and in Vivo Effectiveness in Healing-Impaired Db/Db Mice. J. Controlled Release 2014, 185, 51–61, doi:10.1016/j.jconrel.2014.04.032.
- Gainza, G.; Chu, W.S.; Guy, R.H.; Pedraz, J.L.; Hernandez, R.M.; Delgado-Charro, B.; Igartua, M. Development and in Vitro Evaluation of Lipid Nanoparticle-Based Dressings for Topical Treatment of Chronic Wounds. Int. J. Pharm. 2015, 490, 404–411, doi:10.1016/j.ijpharm.2015.05.075.
Comment 5. Explain why the size of EGF loaded NLC is higher than the blank NLC.
AUTHORS’ RESPONSE: This is an interesting observation. In both cases, we used exactly the same protocol for generating EGF-NLC and blank-NLC. However, it is true that EGF-NLC particle size was higher than blank-NLC. Although the reasons of this difference could not be elucidated in the present work, we may hypothesize that the incorporation of EGF could slightly modify the crystallin structure inside the nanoparticles, and this is known to be related to particle size [3]. This interesting information has been incorporated to the discussion section of the revised manuscript (lines 379-384).
- Sangsen, Y.; Laochai, P.; Chotsathidchai, P.; Wiwattanapatapee, R. Effect of Solid Lipid and Liquid Oil Ratios on Properties of Nanostructured Lipid Carriers for Oral Curcumin Delivery. Adv. Mater. Res. 2015, 1060, 62–65, doi:10.4028/www.scientific.net/AMR.1060.62.
Comment 6. Some sentences are too long and difficult to follow i.e. lines 378-381, please check carefully the language.
AUTHORS’ RESPONSE: The manuscript has been carefully revised and checked by a professional native English-speaker editor, as suggested.